# Order Matters in the Presence of Dataset Imbalance for Multilingual Learning

**Dami Choi**[*][†]
U. Toronto & Vector Institute
choidami@cs.toronto.edu

**Derrick Xin**[*]
Google Research
dxin@google.com

**Hamid Dadkhahi**
Google Research
hdadkhahi@google.com

**Justin Gilmer**
Google Deepmind
gilmer@google.com

**Ankush Garg**
Google Deepmind
ankugarg@google.com

**Orhan Firat**
Google Deepmind
orhanf@google.com

**Chih-Kuan Yeh**
Google Deepmind
chihkuanyeh@google.com

**Andrew M. Dai**
Google Deepmind
adai@google.com

**Behrooz Ghorbani**
OpenAI
ghorbani@openai.com

## Abstract

In this paper, we empirically study the optimization dynamics of multi-task learning, particularly focusing on those that govern a collection of tasks with significant data imbalance. We present a simple yet effective method of pre-training on high-resource tasks, followed by fine-tuning on a mixture of high/low-resource tasks. We provide a thorough empirical study and analysis of this method's benefits showing that it achieves consistent improvements relative to the performance trade-off profile of standard static weighting. We analyze under what data regimes this method is applicable and show its improvements empirically in neural machine translation (NMT) and multi-lingual language modeling.

## 1  Introduction

Over the past few years, large multi-task neural networks have emerged as a popular modeling paradigm in deep learning. The appeal behind these models is that they can leverage transfer learning among the tasks to outperform single-task models. Indeed, multi-task models have achieved state-of-the-art performance in domains such as machine translation [2, 8], language understanding [24, 32], and speech recognition [4, 3].

Unfortunately, optimizing such multi-task models remains a challenge. To effectively train these models, the different tasks need to be balanced during training. This is often done by sampling each task with a static probability.

Prior work [31, 20] shows evidence that when all tasks are in the data rich regime (high-resource), such static sampling approaches yield optimal results. However, when certain tasks are data sparse

---

[*]Equal contribution [†]Work done as a student researcher at Google.

37th Conference on Neural Information Processing Systems (NeurIPS 2023).

(low-resource)[2], which is quite common in real-world applications, the optimality of static sampling is unclear.

The problem with static sampling in the presence of low-resource tasks is that it has difficulty dealing with overfitting on the low-resource tasks. This is because early stopping is not a viable solution due to high-resource tasks needing many more epochs to converge. The transfer learning scheme of pre-training on high-resource and fine-tuning on low-resource tasks (such as in [33]) provides a solution to the overfitting problem, since the training of high and low-resource tasks are separated. Not only this, but the training of low-resource tasks can potentially benefit from positive transfer that comes from performing well on the high-resource tasks. The problem with this approach, however, is that during the fine-tuning phase, catastrophic forgetting of the pre-training tasks ensues.

In this paper, we introduce a simple training scheme that combines the best of static sampling and transfer learning: pre-train on a high-resource task and fine-tune jointly on a mixture of high and low-resource tasks. A pre-training and fine-tuning scheme effectively enables early stopping by allowing the training of low-resource tasks to happen for as little as needed to prevent overfitting, while training the high-resource task for as long as needed. Furthermore, pre-training on a high-resource task will potentially enable positive transfer for low-resource tasks and result in faster convergence in the fine-tuning phase. Lastly, the fine-tuning phase on a mixture of high and low-resource tasks will not only remedy the catastrophic forgetting issue of fine-tuning only on low-resource tasks, but also enjoy further transfer learning among all the tasks.

Through an extensive empirical study, we find that the pre-training and joint fine-tuning scheme yields superior low-resource task performance compared to both static sampling and the transfer-learning scheme. We observed that the performance improvement on static sampling is driven by two mechanisms. The first is that pre-training initializes the fine-tuning phase at a better starting point than random initialization due to positive transfer. The second is that higher sampling rates are more data-efficient than lower sampling rates. Because our method has two separate training phases, the low-resource-training phase can be short. This in turn enables us to increase the low-resource sampling rate without risking overfitting. Indeed, our method is more data-efficient than static sampling in terms of the low-resource tasks throughout the entire fine-tuning phase, achieving better low-resource task performance while using only a fraction of the data seen by static sampling. We further observe that pre-training and joint fine-tuning seems to have a regularization effect. However, we find that regularization is not the main factor behind the performance improvement, since increased explicit regularization, such as dropout, does not improve the performance to the extent that our method does.

The contributions of this paper can be summarized as follows:

- To the best of our knowledge, we are the first to show that it is possible to push the Pareto front of static sampling in the data-imbalanced regime.
- We present a simple algorithm that can be readily used to boost low-resource tasks' performance in multilingual models.
- We show on realistic workloads (up to 13B parameters) that our scheme performs better than static sampling and transfer learning with respect to the low-resource language-pair/language.

## 2   Background

In our work, we focus on the supervised setting, where our model parameters $\theta \in \mathbb{R}^p$ are trained on $K$ different tasks, with the loss for task $i$ being $\mathcal{L}_i(\theta)$.

We introduce the idea of Pareto optimality to better explain the trade-off effect that happens when training on many different tasks.

**Definition** (Pareto Optimality). *$\theta \in \mathbb{R}^p$ Pareto dominates another $\theta'$ if $\forall 1 \leq i \leq K$, $\mathcal{L}_i(\theta) \leq \mathcal{L}_i(\theta')$ and there exists a task $j$ where $\mathcal{L}_j(\theta) < \mathcal{L}_j(\theta')$. $\theta$ is Pareto optimal if it is not dominated by any other point. The collection of the Pareto optimal points is denoted as the Pareto front.*

---

[2]In this literature, data rich and data sparse tasks are often referred to as high-resource and low-resource respectively. Note that whether a task is high-resource or not depends on both the amount of training data and the model capacity.

A standard approach for optimizing multi-task models is *scalarization* [5] or static sampling:

$$\hat{\boldsymbol{\theta}}(\boldsymbol{w}) = \arg\min_{\boldsymbol{\theta}} \sum_{i=1}^{K} \boldsymbol{w}_i \mathcal{L}_i(\boldsymbol{\theta}), \tag{1}$$

where $\boldsymbol{w}$ is a fixed vector of pre-determined task weights with $\boldsymbol{w} > 0$ and $\sum_i \boldsymbol{w}_i = 1$.

In our work, we follow convention and implement scalarization via proportional sampling, where data from task $i$ is sampled with probability equal to $\boldsymbol{w}_i$. In this case, the expected loss is equal to the loss from scalarization:

$$\mathcal{L}(\boldsymbol{\theta}) = \mathbb{E}_{\boldsymbol{x}}\left[\ell(\boldsymbol{x};\boldsymbol{\theta})\right] = \sum_{i=1}^{K} \mathbb{P}(\text{task } i)\mathbb{E}_{\boldsymbol{x}\sim\text{task } i}\left[\ell(\boldsymbol{x};\boldsymbol{\theta})\right] = \sum_{i=1}^{K} \boldsymbol{w}_i \mathcal{L}_i(\boldsymbol{\theta}). \tag{2}$$

Prior work [31] studied the performance trade-off behavior of scalarization and a variety of different multi-task optimization (MTO) methods in the two-task setting. They found that both in the high-resource case and in the data-imbalanced case, no MTO method improved upon the Pareto front of scalarization. In our work, we compare the performance trade-off behavior of scalarization and our proposed method, and find that the Pareto front of scalarization can be improved in the data-imbalanced regime.

Note that practically speaking, it is not feasible to determine whether $\boldsymbol{\theta}$ is truly Pareto optimal since we must check that it is not dominated by all $\boldsymbol{\theta}' \in \mathbb{R}^p$. Following [31], instead of considering all of $\mathbb{R}^p$ we consider only the parameters reachable by a fixed set of hyperparameters.

# 3   Pre-training Joint Fine-tuning

Given $K$ tasks, among which some are low-resource, our goal is to optimize the performance of the low-resource tasks without sacrificing the performance of the remaining tasks. Static sampling is not ideal because all tasks are seen constantly throughout the entirety of training, resulting in overfitting of low-resource tasks while high-resource tasks still need to be learned. Naively breaking up training into two phases and training on low-resource tasks in the later phase results in catastrophic forgetting of earlier-trained tasks.

Assuming the existence of at least one high-resource task, we propose to first pre-train on a high-resource task, and fine-tune the resulting model on the full mixture of $K$ tasks. We call this method **pre-training joint fine-tuning**[3].

In our preliminary experiments, we found that it is important to reset the learning rate schedule and optimizer state when switching over to the joint fine-tuning phase. This is because learning is extremely slow for tasks that are newly introduced when the learning rate has already decayed. In our evaluations, we additionally experiment with adding resetting to the scalarization baseline to ensure that improvements from our method are not purely from resetting. See Sections 4.1.2 and 4.2 for more detail.

Our two-stage training process introduces additional hyperparameters compared to scalarization: the hyperparameters involved in the pre-training phase, and the length of the pre-training phase. However, we find that tuning is not much more difficult than scalarization, and in some cases it is easier to tune. The pre-training phase only involves tuning for a single task, which is much easier than tuning for multiple tasks. We also expect the joint fine-tuning phase to be shorter than the full training length of scalarization; therefore, tuning for the second phase should be around the same or easier than scalarization. Lastly, our results show that pre-training does not hurt fine-tuning performance and longer pre-training translates to better fine-tuning. From this, we recommend that if there is a strict training budget, it is better to be conservative and pre-train for a shorter amount of time. However, if the goal is to obtain the best performance and there is no strict compute budget, we recommend pre-training for as long as possible before fine-tuning. See Section 4.3 for more details.

---

[3]We use the terms 'pre-training' and 'fine-tuning' only to distinguish the two phases of training, and that the training objectives are the same for both phases. In other words, we do not suggest using any particular self-supervised objective for the pre-training phase, or training on downstream tasks for the fine-tuning phase.

# 4 Experiments

In the following sections, we apply our proposed training scheme to NMT (where each task is a language-pair) and multilingual training (where each task is a language). In the NMT experiments, we show that pre-training joint fine-tuning pushes past the trade-off frontier of scalarization through significant improvements on the low-resource task– a feat that many popular gradient-based multi-task optimization methods were not able to achieve [31]. In the language modeling experiments, we scale up the number of tasks, and show that our method retains the same benefits for the low-resource languages.

## 4.1 Neural Machine Translation

For our first experiment, we focus on a setting where we can trace out, and compare the trade-off frontiers obtained with and without pre-training. As in prior work [31], we choose to work on the two-task setting due to the ease of visualizing the performance trade-off curves.

We choose our high and low-resource language-pairs from the WMT dataset, where English→{Chinese, French} are the high-resource language pairs, and English→{Romanian, Hindi} are the low-resource language pairs. See Table 1 for details on each language-pair. All models in this section use a pre-LayerNorm encoder-decoder transformer architecture [28]. In the main paper, we present results on models with three encoder layers and three decoder layers. Results obtained with a larger model size are in Appendix A.2. Further details, including hyperparameters, are in A.1.

In order to trace out the trade-off frontiers for the pre-training joint fine-tuning method and the scalarization baseline, we adhere to the following methodology. For scalarization, we iterate through a grid of task weights (since there are only two tasks, a grid is a linear function of the granularity) and train on the two language pairs for $N$ steps using proportional sampling according to the task weights. For the pre-training joint fine-tuning method, we first pre-train on the high-resource language pair for $N_1$ training

Table 1: Overview of data sources used in our NMT experiments. Our datasets are from WMT.

| Language Pair | # Train Ex. | # Eval Ex. |
|---|---|---|
| En-Fr '15 | $40,853,298$ | $4,503$ |
| En-Zh '19 | $25,986,436$ | $3,981$ |
| En-Ro '16 | $610,320$ | $1,999$ |
| En-Hi '14 | $313,748$ | $520$ |

steps. We then reset the optimizer state and the learning rate schedule and fine-tune on a mixture of high-resource and low-resource language pairs for $N_2$ training steps such that $N_1 + N_2 = N$. For the fine-tuning phase, we iterate through a grid of task weights as with scalarization. The grid of sampling rates will trace a performance trade-off front, which can be used to compare our method and scalarization.

Lastly, we train a *restart baseline* in order to ablate the possibility that any improvements coming from pre-training joint fine-tuning are due to the resetting of optimizer state and learning rate schedules before fine-tuning. The restart baseline takes the model obtained via scalarization trained for $N_1$ steps, resets optimizer states and the learning rate schedule, and continues to train it with the same sampling rate as in scalarization.

### 4.1.1 High-Resource and High-Resource:

We first start by highlighting that pre-training joint fine-tuning does not show benefits if all tasks are high-resource. Figure 1 shows that in the English→{Chinese, French} translation tasks, the performance on each of the language-pairs are bounded by the amount of data seen from that pair. In other words, pre-training on En→Fr cannot act as a proxy for En→Zh data, because if it could, the front would be improved. At the same time, pre-training does not negatively impact En→Zh training. Figures 21 and 22 show that pre-training does not affect the learning efficiency for En→Zh (slope of the curves are similar to one another), and also did not result in a worse initialization for En→Zh.

### 4.1.2 High-Resource and Low-Resource

In the data-imbalanced setting of English→{Romanian, French}, we pre-train for 400k steps and fine-tune for 50k steps to emphasize the computational benefits of pre-training fine-tuning. Although a single full run of scalarization ($N$ steps) and pre-training fine-tuning ($N_1 + N_2 = N$) take the same

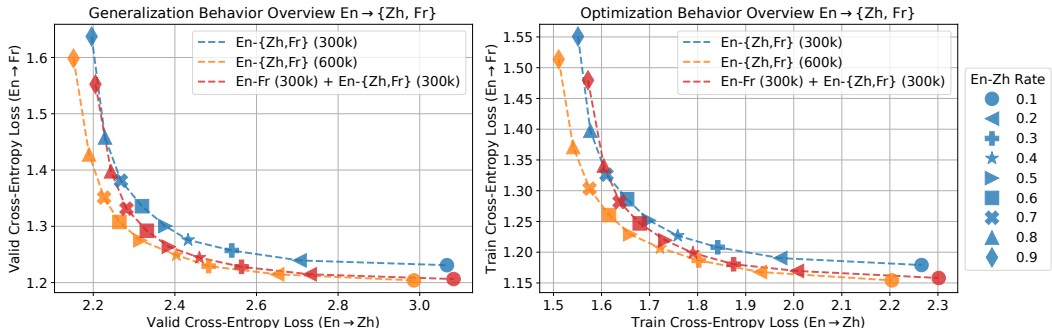

Figure 1: The trade-off front from pre-training does not improve upon the trade-off front from fully static sampling when all tasks are high-resource. The performance on each of the high-resource tasks are bounded by the amount of data seen for that task. We can also observe interference between the two tasks from how all 9 different sampling rates form the trade-off frontier. These observations hold for both testing (*left*) and training (*right*).

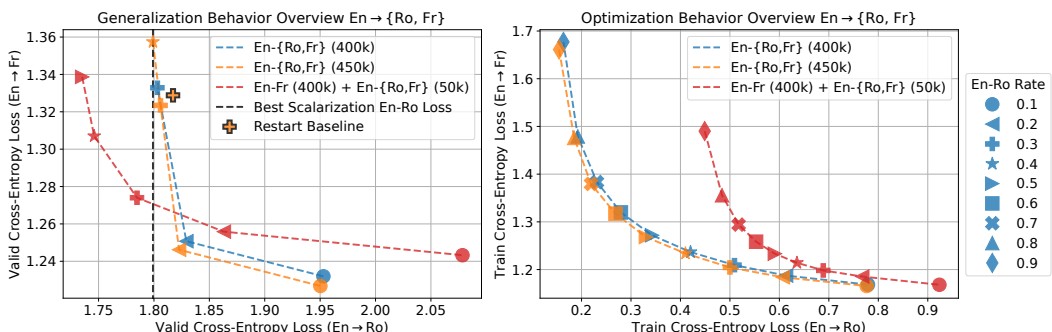

Figure 2: (*Left:*) In the data-imbalanced case, the trade-off front from pre-training yields better low-resource task performance than the trade-off front of scalarization. The poor performance of the restart baseline shows that the resetting of states is not why pre-training and fine-tuning performs well. Note that the trade-off fronts consist of only a subset of the sampling ratios due to overfitting, which is different from the fully high-resource setting. *Right:* Pre-training results in a noticeably worse performance on the training set, hinting that pre-training has a regularization effect on the low-resource task.

amount of compute, pre-training joint fine-tuning makes hyperparamter tuning much more efficient, since 1) tuning for pre-training is on a single task and therefore, easier to tune, and 2) tuning for fine-tuning is faster since $N_2 \ll N$.

In Figure 2 we can observe that pre-training joint fine-tuning is able to achieve performance trade-off points that go beyond what is achievable via scalarization. Pre-training on a high-resource language pair creates non-dominated points by yielding significantly better performance in the low-resource task (En→Ro) without completely sacrificing performance in the high-resource task (En→Fr). Additionally, it is able to do this while seeing less overall Romanian tokens according to Figure 3.

We see similar results for En→{Hi, Fr}, shown in Figure 12 in the Appendix. This is a surprising result since French and Hindi are less linguistically similar than French and Romanian. Finally, we can see from the sub-optimal performance of the restart baseline in Figures 2 and 12 that the act of resetting is not the reason behind the success of the pre-training joint fine-tuning scheme. We provide BLEU score evaluations for En→{Ro, Fr} and En→{Hi, Fr} in Appendix A.5, validating that the improvements in loss translate to downstream metrics.

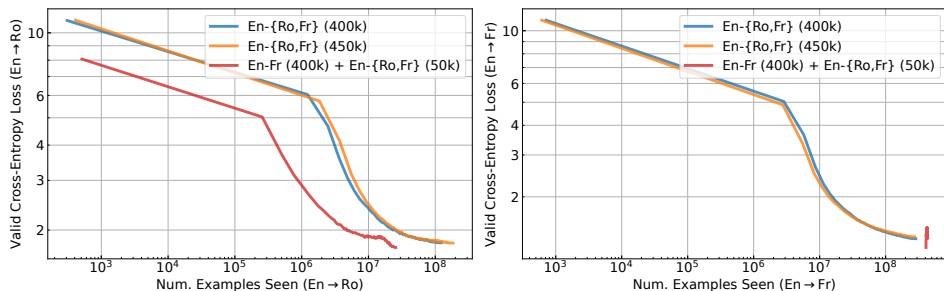

Figure 3: Pre-training joint fine-tuning has both better initialization and data-efficiency than scalarization. Each line corresponds to the datapoint that achieved the best En→Ro validation loss in Figure 2 among the different run groups.

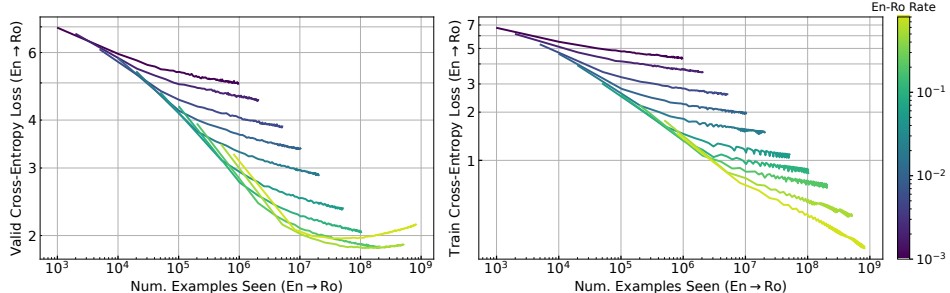

Figure 4: Each curve corresponds to a single scalarization trial with a particular (static) sampling rate for En→Ro. The rate at which the training loss decreases is slower for lower En→Ro sampling rates than for higher sampling rates. At higher sampling rates, overfitting starts to happen.

### 4.1.3 Analysis

The performance improvement of pre-training joint fine-tuning stems from two main mechanisms.

- Pre-training utilizes positive transfer between tasks, and initializes the fine-tuning phase at a better starting point than random initialization. Figure 3 shows this effect for the En→{Ro, Fr} translation tasks.
- Higher sampling rates are more data-efficient than lower sampling rates. Figure 4 shows how optimization (training set performance) gets more and more data-efficient as the sampling rate increases. However, on the generalization side, increasing the sampling rate works only up until a certain point, where overfitting kicks in.

By design, pre-training joint fine-tuning has two separate training phases which allows the low-resource-training phase to be short. This in turn enables us to increase the low-resource sampling rate, resulting in faster training. This effect can be seen in Figure 2, where the En→Ro sampling rates that resulted in the best En→Ro performance was 0.4, while for pre-training joint fine-tuning, the best rate is 0.5. Figure 3 confirms that indeed after pre-training, fine-tuning on En→Ro is more data-efficient than not pre-training.

Joint fine-tuning is also an important piece in addition to the two-stage setup. Only fine-tuning on the low-resource task, which is the classic transfer learning scheme, results in overfitting and catastrophic forgetting of the pre-training task as shown in Figure 6.

Lastly, Figure 2 shows that pre-training joint fine-tuning yields worse training set performance, and therefore, could be seen as having a regularization effect. We show in Figure 5 that regularization by itself does not explain the superior performance of our scheme.

The results seen so far show that data order matters when training in the presence of a low-resource task, since seeing high-resource data first before seeing low-resource data later pushes the pareto front of seeing both types of data at the same time.

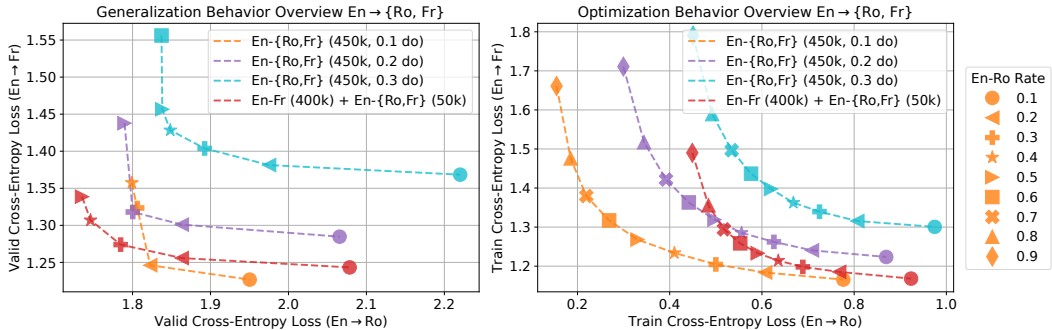

Figure 5: pre-training joint fine-tuning has a regularization effect, but cannot be replaced by simply increasing regularization strength. The dropout rate used in pre-training joint fine-tuning is 0.1.

## 4.2 Multilingual Training

In this section, we expand from a two-task setting to a many-task setting. We train on five languages from the mC4 dataset [32]–English, Hindi, Gujarati, Swahili, and Gaelic– using the span corruption objective from T5 [24]. See Table 2 for details on the dataset. Canonically the mC4 dataset is used in the pre-training phase for models (not to be confused by our pre-training joint fine-tuning method). These models are subsequently applied to downstream tasks such as question answering. This multilingual pre-training phase is also known as the language balancing problem. Our goal is to show that our two stage method can effectively balance high-resource and low-resource languages, improving performance on low-resource languages beyond what is achievable by the conventional method of temperature sampling while not sacrificing performance on high-resource languages.

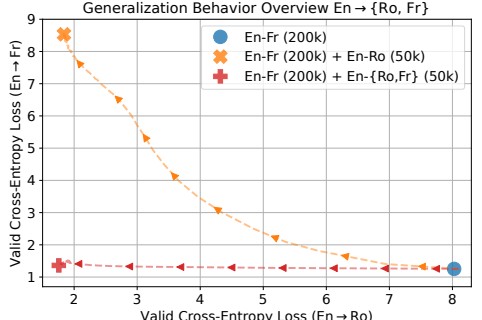

Figure 6: Fine-tuning solely on the low-resource task (En→Ro) leads to both catastrophic forgetting of the pre-trained task (En→Fr) and worse low-resource task performance than fine-tuning on all tasks (En→{Ro, Fr}).

Note that in the mC4 corpus, English is $16745$ times larger than the smallest language we use. This data imbalance underscores the necessity for effective language balancing, particularly in determining the proportion of each language to be used during training. This presents a highly challenging and computationally demanding problem, as it is not feasible to simply sweep the scalarization weights as one would in a two-task setting.

Table 2: Data used from mC4.

| Language | # Chars (B) |
|---|---|
| En (English) | $13,396$ |
| Hi (Hindi) | 75 |
| Gu (Gujarati) | 3.6 |
| Gd (Gaelic) | 0.8 |
| Sw (Swahili) | 4.1 |

For our training setup we closely follow mT5 [32] for the model architecture and training procedure. Specifically, we use the mT5-XXL model (13B parameters), which is an encoder-decoder transformer architecture. Additional training details are available in Appendix B.

**Temperature Sampling** Because we increase the amount of tasks in this setting, detailing the full scalarization trade-off frontier would be computationally infeasible. Therefore, we employ the widely used *temperature sampling* heuristic [11, 7, 2]. Let $D_i$ be data size of language or task $i$, we then define the empirical distribution $\mathbb{P}$ for each task $i$ as:

$$\mathbb{P}(\boldsymbol{x} \in \text{task } i) = \frac{D_i}{\sum_j D_j}. \qquad (3)$$

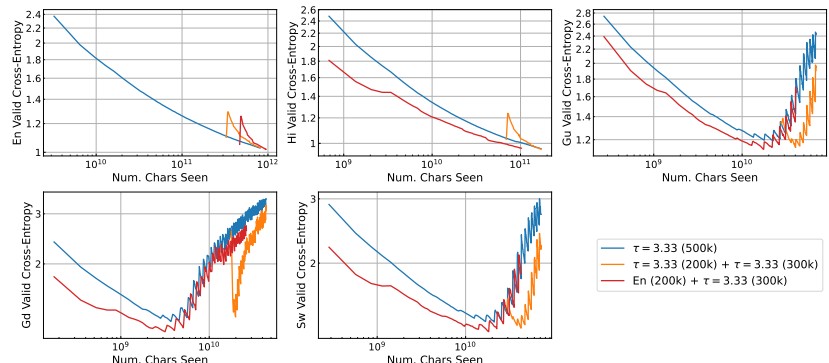

Figure 8: Pre-training on English and joint fine-tuning on all 5 languages leads to better optima for Gujarati, Gaelic and Swahili, the 3 low-resource languages. Pre-training also results in better initialization and token-efficiency for all languages newly seen in the fine-tuning phase.

Temperature sampling then uses a distribution $\mathbb{Q}$ defined by a temperature parameter $\tau$ as follows:

$$\mathbb{Q}(\boldsymbol{x} \in \text{task } i) = \frac{\mathbb{P}(\boldsymbol{x} \in \text{task } i)^{1/\tau}}{\sum_j \mathbb{P}(\boldsymbol{x} \in \text{task } j)^{1/\tau}} \tag{4}$$

The temperature parameter $\tau$ controls the peakiness (or flatness) of the sampling distribution. Commonly used $\tau$'s in the literature are greater than 1, which essentially up-samples low-resource tasks and down-samples high-resource tasks.

**Static Sampling Baseline**   Temperature sampling is ubiquitous due to its simplicity and intuitiveness, but its performance varies greatly with $\tau$. For our static sampling baseline, we tuned $\tau$ among commonly used values in the literature (1.43, 2, 3.33, 5) at a smaller scale, and found that $\tau = 3.33$ performed the best in terms of low-resource languages. We also tried a more intricate sampling strategy called Uni-Max [6], but found that on the 5 languages we chose, it did not perform better than $\tau = 3.33$.

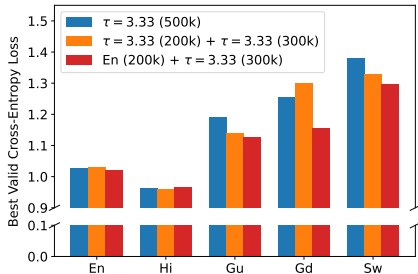

Figure 7: Pre-training joint fine-tuning yields the best performance in 4 out of 5 languages, with significant improvements in the low-resource tasks.

**Pre-training joint Fine-tuning**   For our pre-training joint fine-tuning setup, we first pre-train on English, reset the optimizer state and learning rate schedule, and then fine-tune on all 5 languages using temperature sampling. We use the same sampling rates as the static sampling baseline ($\tau = 3.33$) to reduce the tuning overhead over static sampling.

As in the NMT experiments, we employ a restart baseline to fully ablate the pre-training fine-tuning scheme. The restart baseline resets the optimizer state and learning rate schedule in the middle of training for the static sampling baseline.

**Results**   Figures 7 and 8 show that while a learning rate schedule restart helps performance, pre-training joint fine-tuning yields the best results on the low-resource tasks. Surprisingly, it not only improves the performance on Gujarati, Gaelic, and Swahili, but also shows a slight enhancement on English. We note that due to the vast dataset imbalance, the temperature sampling baseline overfits on the low-resource tasks before English has a chance to converge. Consequently, pre-training joint fine-tuning can leverage the benefits mentioned in the previous section–regularization, transfer, and reduced forgetting–to achieve a superior lower bound performance with higher token efficiency.

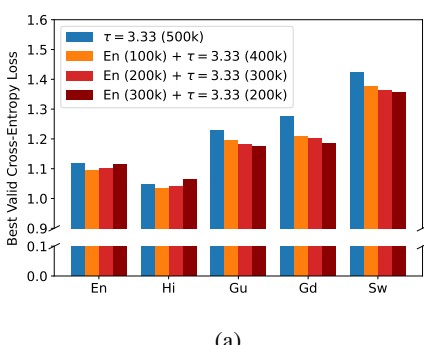 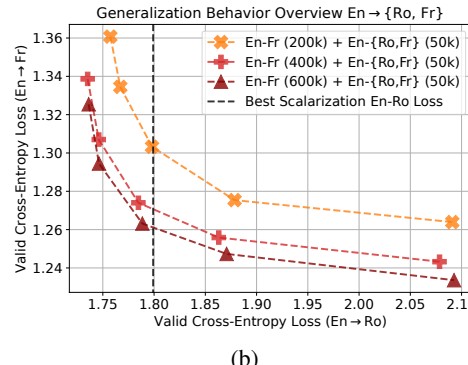

(a)                                             (b)

Figure 9: *Left*: For language modeling on mC4, longer pre-training leads to better best-achievable performance for the 3 low-resource languages (Gu, Gd, Sw) despite the decreased length of fine-tuning. On the other hand, due to the decreased length of fine-tuning, high-resource languages do not enjoy the benefits of pre-training. *Right*: For NMT, when the training budget is not fixed, longer pre-training leads to better overall performance trade-off fronts.

## 4.3 Length of Pre-training

Our method is simple but comes with some choices to make, one of which is the number of steps to pre-train for. We investigate the effect of the number of pre-training steps in NMT and language modeling on mC4 by pre-training with less, and more steps than in the previous sections. With the language modeling task, we fix the total training length to be 500k steps to emulate a compute-constrained scenario. We chose to use a smaller model (mT5-XL as opposed to mT5-XXL used in Section 4.2 for faster training). With NMT, we fix the number of fine-tuning steps, but let the total training steps vary.

Figure 9(a) displays the effects of varying pre-training length in the mC4 experiments. We see that longer pre-training improves best achievable performance on the low-resource tasks of Gujarati, Gaelic, and Swahili. This is despite the fact that the number of fine-tuning steps decreased due to the fixed total step budget. In other words, for the 3 low-resource tasks, longer pre-training improves performance more than exposure to the tokens. On the other hand, performance on English and Hindi worsens with increased pre-training length. For English, this is due to the resetting of the learning rate schedule and the decreasing of fine-tuning steps. Resetting involves a learning rate warmup, which worsens English performance before improving again (see the panel corresponding to En for Figure 8). Decreasing fine-tuning steps gives English less time to recover its performance from pre-training. For Hindi, the worsened performance is simply because it is not a low-resource task in this context, and therefore, less tokens seen translates to worse performance.

In Figure 9(b) we see that in the NMT experiments, pre-training longer on En→Fr translates to better overall trade-off fronts, not just for the low-resource task.

The implications of these results are that when there is a strict training budget, it is better to be conservative and pre-train for a shorter amount of time. However, if the goal is to obtain the best performance with no strict compute budget, it is better to pre-train for as long as possible before fine-tuning. Note that longer overall training is an option for our method (by pre-training for longer) but not for static sampling because static sampling needs to constantly be training on the low-resource tasks, which will lead to overfitting when training for too long.

## 5 Related Work

**Multitask Learning**   Multitask learning has gained increased attention in being able to learn many tasks in an efficient way due to parameter sharing and transfer between tasks. In the language domain, multilingual neural machine translation [12, 14] enables translation from multiple source languages to multiple target languages. Due to the transfer of information between language pairs, multilingual NMT has seen improvements in low-resource language-pair performance compared to training solely

on that language pair [12]. In addition to NMT, large multilingual pre-trained language models are used to fine-tune on a variety of downstream tasks with different languages [32]. Prior works on intermediate training take advantage of cross-task [23] and cross-lingual [22] transfer to improve downstream task performance. However, in multilingual approaches there exists the problem of dataset imbalance, where low-resource languages tend to suffer in performance. Recently, [6] found that naive temperature sampling might lead to overfitting of low-count languages, and suggested epoch capping with a uniform distribution for high-count languages, showing improvements over temperature sampling. In multilingual NMT, to our knowledge, we are the first to show that a simple pre-training stage on a high-resource language pair can improve the trade-off front of static sampling. Furthermore, our method is orthogonal to innovations in sampling strategies like [6], and can potentially show better results in conjunction with better sampling.

**Transfer Learning in NMT**    The benefits of transfer learning to low-resource language-pairs has been long known in the NMT literature [33, 9, 17]. [33] showed that pre-training on a high-resource language pair can improve performance compared to training from scratch. While most prior work on transfer learning in NMT focus on improving performance on low-resource bilingual data, recent work [21] used transfer learning to improve performance on multiple language pairs. Unlike the transfer learning literature in NMT [21, 15], we show that pre-training can push the low-resource frontier in the multilingual setting, by testing a grid of sampling rates and hyperparameters to trace the trade-off front. Prior work in the literature study the relationship between the pre-training and fine-tuning language pairs [10], freezing different parts of the model during fine-tuning [1], and experimenting with many-stage pre-training [9]. We expect to further benefit from research done in this direction.

**Curriculum Learning**    Due to the imbalanced nature of multilingual datasets, a static sampling strategy is unsatisfactory. [30] used a hand-crafted temperature sampling schedule that samples more high-resource earlier in the training, and gradually samples more low-resource languages. The performance boost from using such a schedule, compared to a static one, supports our observations from pre-training using a high-resource language pair. On the other hand, there are many works that employ a more intricate strategy for an adaptive schedule [13, 29, 18]. In comparison, our method is simple with little to no overhead. We include discussion on our experience, though preliminary, with trying an adaptive schedule in Appendix C. Lastly, [26] showed that the ordering of data within a task affects catastrophic forgetting, which supports our observations.

## 6    Limitations and Future work

In our experiments, we focus on training on a single high-resource task during the pre-training phase. It would be interesting future work to study pre-training with more than one language or language-pair. We also only experiment with fine-tuning all parameters of the pre-trained model. Studying the effect of freezing different parts of the model during fine-tuning, potentially as a function of the relationship between pre-training and fine-tuning tasks, is left to future work.

## 7    Conclusion

In this work, we demonstrated the benefits of a pre-train joint fine-tune setup for multi-objective optimization when there is a mixture of high and low-resource tasks. We show that in the presence of large data imbalance, the order at which tasks are introduced has significant impact on overall performance. We demonstrate through a variety of experimental settings that this methodology produces points that can go past the trade-off frontier achieved by scalarization. We show that a major weak point of scalarization in this regime is that it overfits on the low-resource task, being unable to early stop due to the high-resource task not converging. Our method both allows the high-resource task to converge during pre-training and prevents overfitting through joint fine-tuning. It also outperforms scalarization that under-samples the low-resource task due to higher token efficiency. We also show that fine-tuning only on the low-resource task, a popular scheme in the NMT literature, is undesirable due to its inability to prevent forgetting. Our method is a simple natural strategy for avoiding the above failure modes. Given the significant performance boost we observe in our experiments, we believe that this training regime has the potential to become a standard approach, particularly in the era of large language models.

## Acknowledgments and Disclosure of Funding

We thank George E. Dahl, Wolfgang Macherey, and Macduff Hughes for their constructive comments on the initial version of this manuscript. Additionally, we thank Sourabh Medapati and Zachary Nado for their help in debugging our code base. Moreover, we are grateful to Soham Ghosh and Mojtaba Seyedhosseini for valuable discussions regarding the role of MTOs in large-scale models. Lastly, we thank Chris J.H. Zhang for helpful discussions.

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

# A  NMT Experiments: Additional Information

## A.1  Detailed Training Setup

This section details the experimental setup used in Section 4.1. We use the pre-LN encoder-decoder transformer architecture. The experiments presented in the main text use three layers for both the encoder and decoder, but we also present results with 6 layers for the encoder and decoder. We follow the convention in NMT literature and train our models with 0.1 label smoothing and 0.1 dropout for feed-forward and attention layers. See Table 3 for complete architecture details.

Table 3: Transformer architecture details and common hyperparameters.

| Hyperparameter | |
| --- | --- |
| Feed-forward dim | 2048 |
| Model dim | 512 |
| Attention heads | 8 |
| Attention QKV dim | 512 |
| Label smoothing | 0.1 |
| Dropout | 0.1 |

We use SentencePiece tokenization [19] to generate a vocabulary of size 64,000 for each NMT problem (e.g. En→{Zh, Fr}).

All models were trained using the Adam [16] optimizer with a batch size of 1024. For all our NMT experiments, we used a linear warmup to the desired learning rate, followed by a cosine decay schedule that decays to 0. This is true for all legs of training for methods that use our scheme; during the pre-training phase, we do a linear warmup followed by a cosine decay, and during the fine-tuning phase, after loading the pre-trained model, we do a linear warmup followed by cosine decay.

For the baseline experiments that do not do pre-training, and also for the pre-training portion, we warmup for 40k steps. For fine-tuning, we tune the warmup steps from within {10k, 20k, 30k, 40k} for all experiments other than for En→{Zh, Fr}, where we warmup for 40k steps. The base number of training steps, and the number of fine-tuning steps are shown in Table 4. Note that for comparison's sake we also trained a baseline-without-pre-training model for 'base + fine-tune' number of steps.

Table 4: Number of training steps for all NMT experiments.

| | 3-layer | | 6-layer | |
| --- | --- | --- | --- | --- |
| | base | fine-tune | base | fine-tune |
| En→{Zh, Fr} | 300k | 300k | 300k | 300k |
| En→{Ro, Fr} | 400k | 50k | 300k | 50k |
| En→{Hi, Fr} | 300k | 50k | 275k | 50k |

For all experiments, we sweep the base learning rate in the grid {2.5e-4, 5e-4, 2.5e-3, 5e-3, 7.5e-3}. We also sweep the sampling rate for En→Fr and En→Cs in the grid $\{i/10\}_{i=1}^{9}$, which fully determines the sampling rate for the other language pair. All plotted points correspond to the **final** measurement taken for each trial.

For all fine-tuning experiments, when loading the pre-trained model checkpoint, we reset the optimizer state. We also trained all parameters of the model, and did not freeze anything.

## A.2 Additional Performance Trade-Off Curves

In this section, we present the performance trade-off curves for En→{Hi, Fr}, as well as for 6-layer models on En→{Zh, Fr}, En→{Ro, Fr}, and En→{Hi, Fr}. The black-bordered points in the generalization portion of Figures 11 and 13 below correspond to the restart baseline.

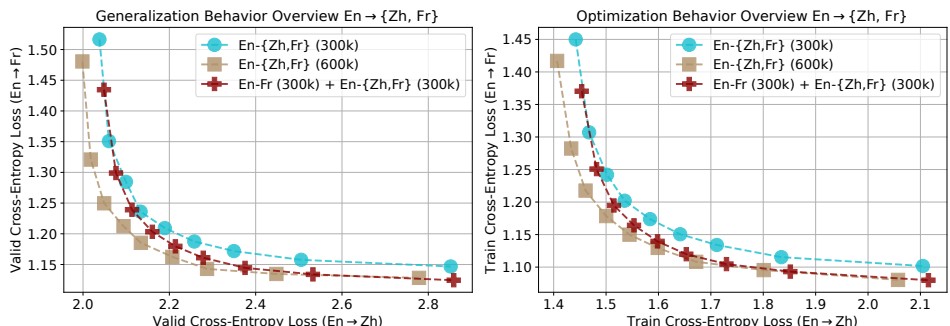

Figure 10: Performance trade-off behavior for En→{Zh, Fr} with 6-layer models. Each point corresponds to the final performance of a model. Similarly to the 3-layer-model case (Figure 1), pre-training does not yield improvements.

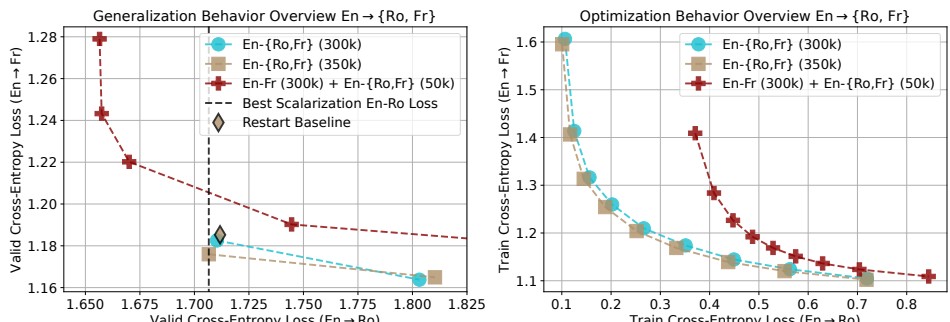

Figure 11: Performance trade-off behavior for En→{Ro, Fr} with 6-layer models. We see a similar behavior as with 3-layer models. In addition, we are able to further improve the performance on both En→Ro due to a larger model size.

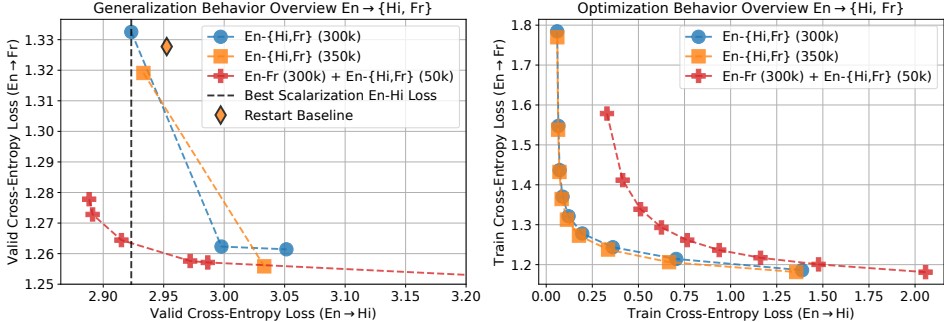

Figure 12: Performance trade-off behavior for En→{Hi, Fr} with 3-layer models. These results mirror those seen in Figure 2. We note that here French and Hindi are more linguistically dissimilar than French and Romanian.

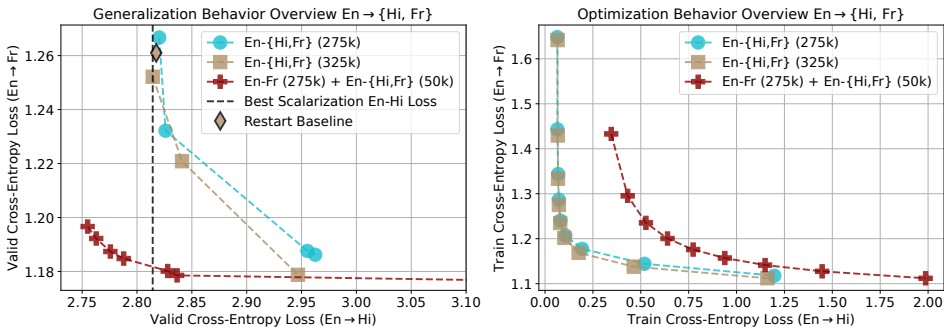

Figure 13: Performance trade-off behavior for En→{Hi, Fr} with 6-layer models. As with the 3-layer models, We observe a similar improvement in both En→Hi and En→Fr performances, despite the dissimilarity of French and Hindi.

## A.3 Performance Trade-Off Curves with Sampling Rate as Markers

In this section, we present the same performance trade-off curves as shown previously, but with the markers representing sampling rates for the lower-resource language pair. We can see that in all but one case (En→{Hi,Fr} 6-layer model; Figure 19), the model that performs the best in the low-resource language pair, samples the low-resource language pair at a higher rate than the baselines that do not use pre-training. The black-bordered points in the generalization portion of Figures 16, 17 18, and 19 below correspond to the restart baseline.

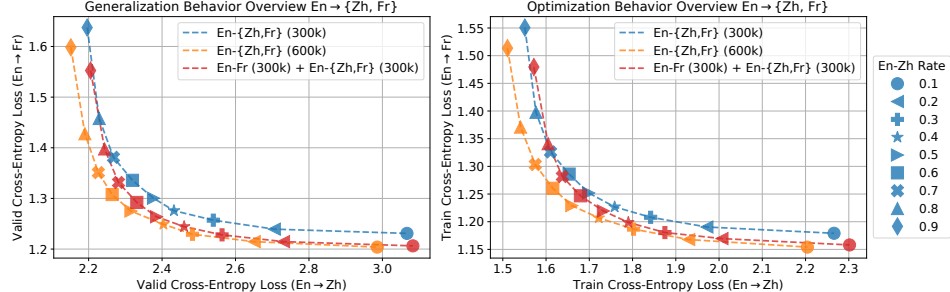

Figure 14: Performance trade-off behavior for En→{Zh, Fr} with 3-layer models. We can clearly see that there is no optimal rate in this case, since we trace a Pareto front as we vary the En→Zh sampling rates from 0.1 to 0.9.

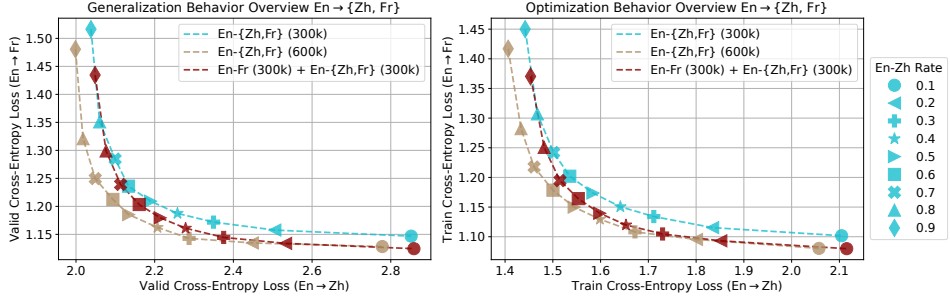

Figure 15: Performance trade-off behavior for En→{Zh, Fr} with 6-layer models. We observe a similar behavior as in the 3-layer case.

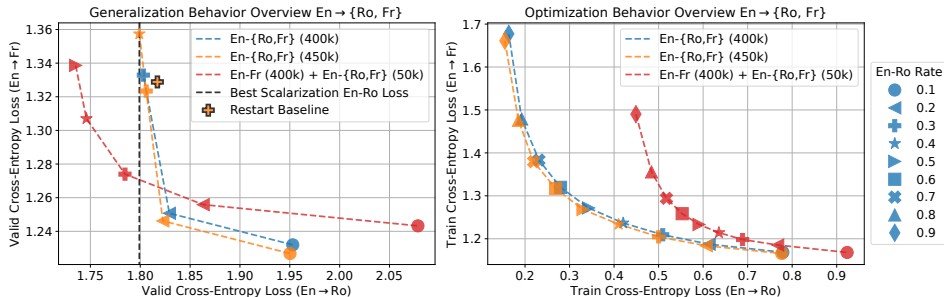

Figure 16: Performance trade-off behavior for En→{Ro, Fr} with 3-layer models. Unlike the En→{Zh, Fr} case, we have a few sampling rates that are more optimal than the rest. Pre-training allows sampling En→Ro at a higher rate without overfitting, than without pre-training.

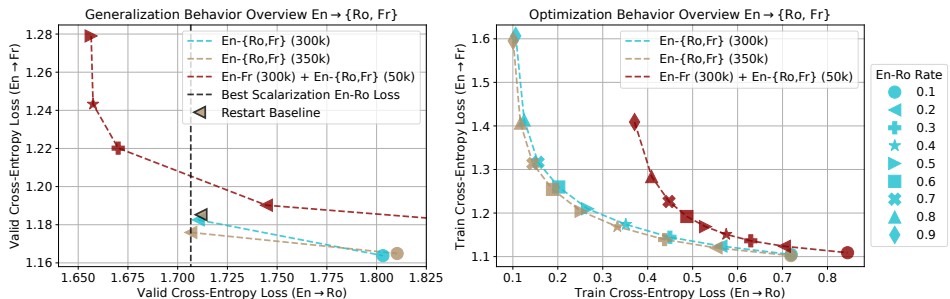

Figure 17: Performance trade-off behavior for En→{Ro, Fr} with 6-layer models. We see a similar behavior as in the 3-layer case.

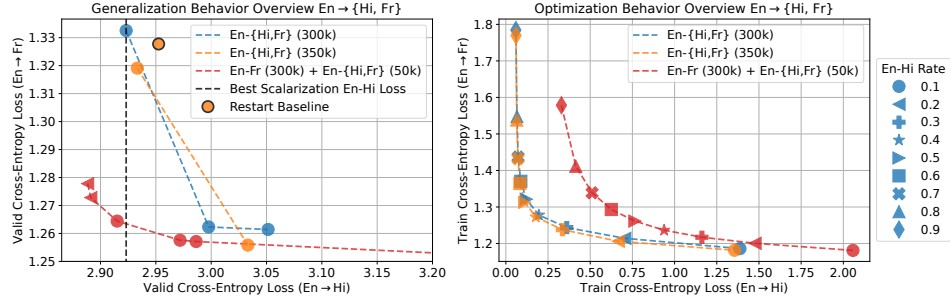

Figure 18: Performance trade-off behavior for En→{Hi, Fr} with 3-layer models. Like in the En→{Ro, Fr}, pre-training allows sampling En→Hi at a higher rate without overfiting than without pre-training.

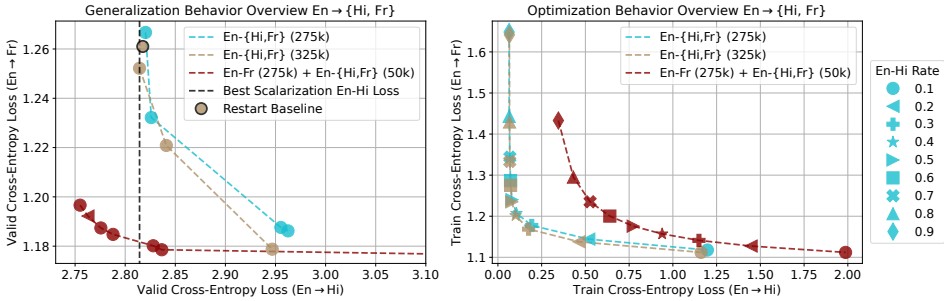

Figure 19: Performance trade-off behavior for En→{Hi, Fr} with 6-layer models. In this case, pre-training still allows sampling En→Hi at a higher rate, but the rate that yielded the best En→Hi was surprisingly the same rate as the baseline without pre-training.

## A.4 Efficiency Plots

In this section, we plot the number of examples seen from one language pair against the validation cross-entropy loss on that language pair. The number of XX→YY examples seen at train step $t$ is computed by multiplying $t$, the batch size, and the sampling rate for XX→YY. Each curve in a given figure corresponds to the trial that achieved the best final validation performance on the low(er)-resource language pair within the method given by the legend (i.e. the blue curve in Figure 20 corresponds to the trial that achieved the best final validation En→Zh cross-entropy loss among all trials that did not use pre-training, and was trained for 300k steps.) For the curves corresponding to our proposed pre-training and fine-tuning scheme, we only show the fine-tuning portion of training.

Note that initial linear decay followed by a smooth decay is an artifact of evaluating on a linear-scale when the plots are in log-scale.

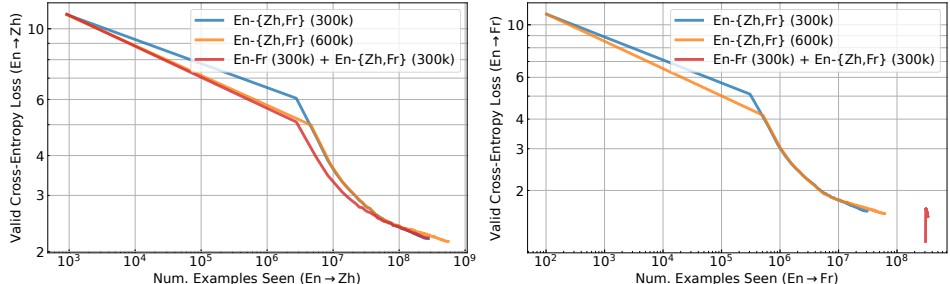

Figure 20: For the 3-layer model, pre-training does not provide any significant gains in training efficiency for En→Zh when pre-training on En→Fr. Given that the blue and red curves coincide towards the end of training, we can anticipate that pre-training did not impair En→Zh training (by providing a suboptimal initialization), and that if we were to train the red curve for 300k more steps, it would be able to catch up with the orange curve (best En→Zh performance).

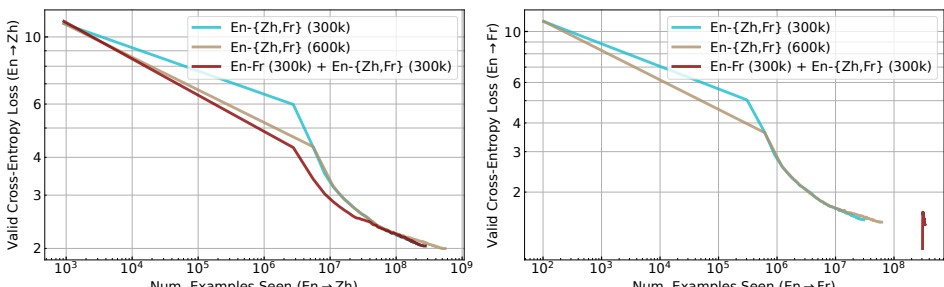

Figure 21: We observe a similar behavior with 6-layer models as with 3-layer models.

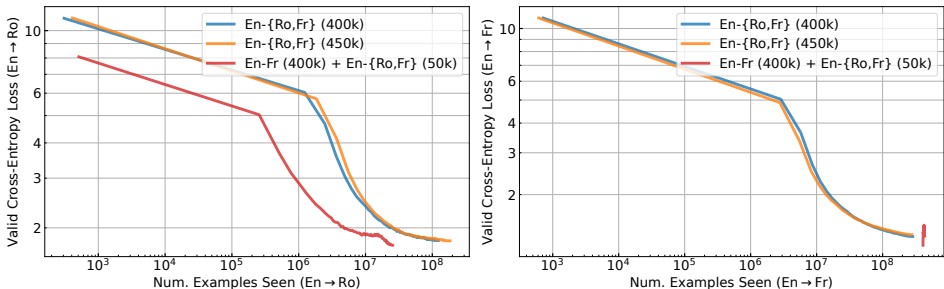

Figure 22: On the 3-layer models, pre-training is able to accelerate training on En→Ro when pre-trained on En→Fr. Even with less overall examples seen in En→Ro, we can perform better than the baselines that did not use pre-training.

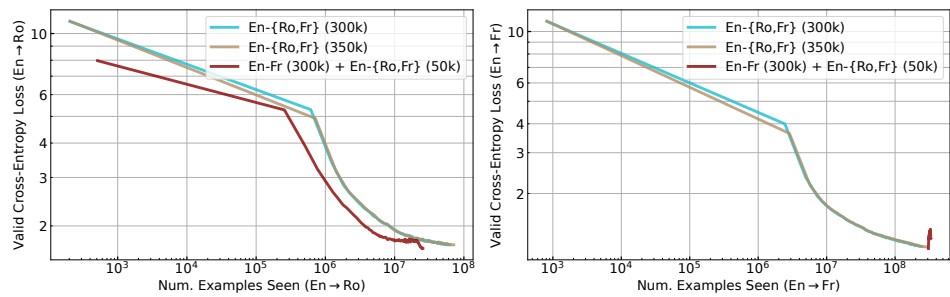

Figure 23: We observe a similar efficiency boost with 6-layer models as with 3-layer models.

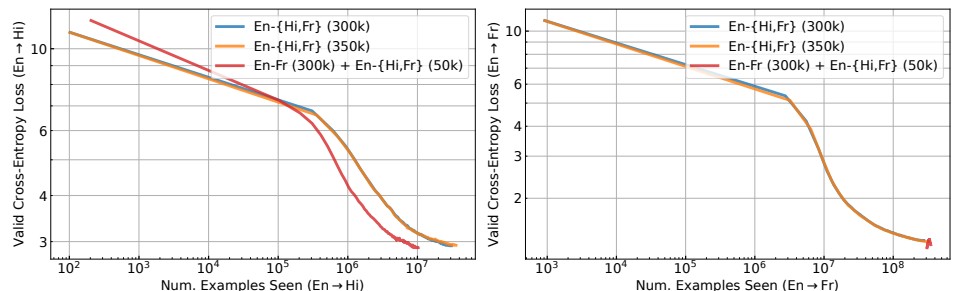

Figure 24: On the 3-layer models, we observe a similar efficiency boost as with En→{Ro,Fr}

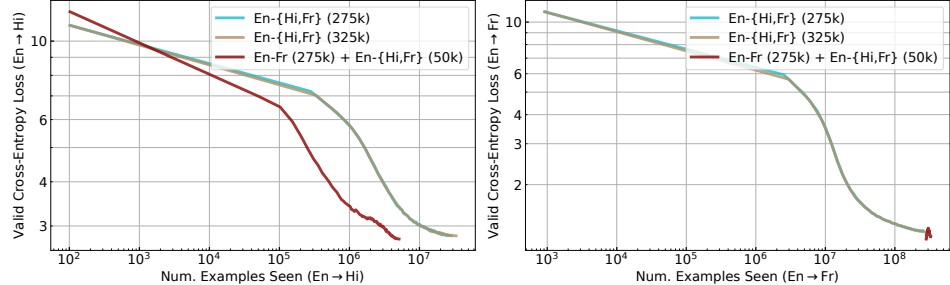

Figure 25: On the 6-layer models, we observe a similar efficiency boost as with 3-layer models.

## A.5    BLEU Score Plots

Here, we present the performance trade-off curves for when the metric is BLEU score instead of cross-entropy loss. All translations are generated via Beam-Search with beam size of 4.

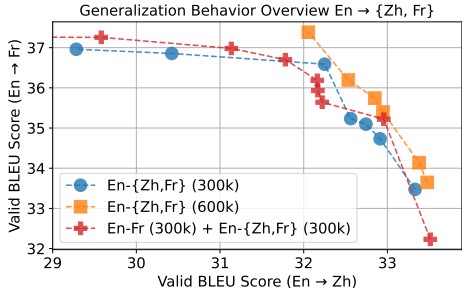

Figure 26: The BLEU score plot paints a better picture for pre-training than the cross-entropy plot (Figure 1), since pre-training was able to improve the En-Zh BLEU score to be on par with the score of joint training for 600k steps. Results are with 3-layer models.

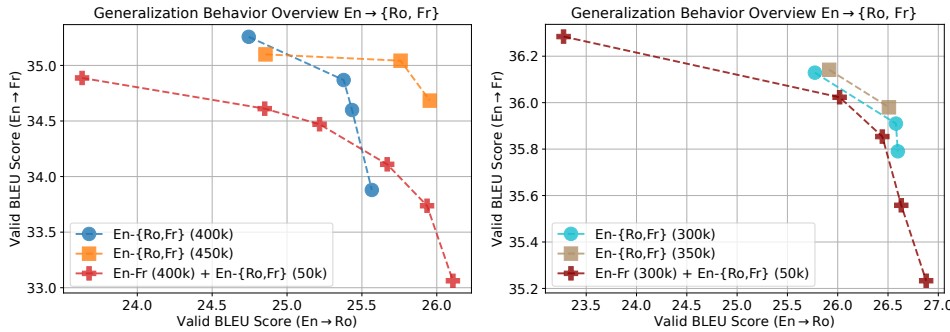

Figure 27: Our proposed pre-training scheme improves upon the best BLEU score for En→Ro without pre-training for both the 3-layer models (*left*) and 6-layer models (*right*).

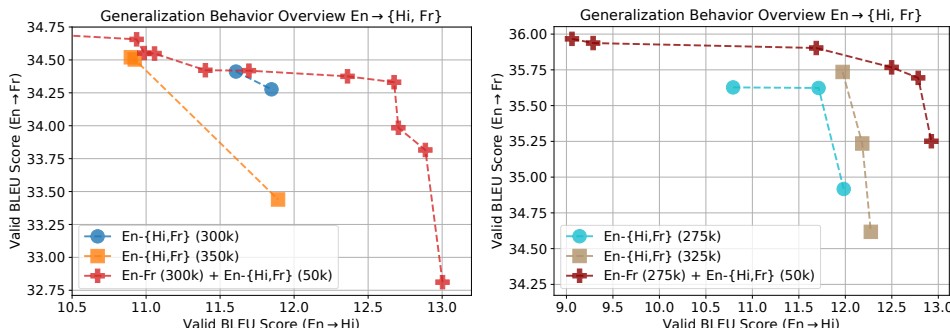

Figure 28: Our proposed pre-training scheme improves upon the best BLEU score for En→Hi without pre-training for both the 3-layer models (*left*) and 6-layer models (*right*). The improvements are more substantial than for En→{Ro, Fr}.

## B    Additional Training Details in Multilingual Training

We use an additionally processed version of the mC4 [32] dataset as proposed in [6] (documents with language ID confidence below 0.95 were filtered).

The model architectures used are the same as mT5 models [32], except that relative position embeddings are not shared across layers. We also use the number of real target tokens as the effective loss normalization instead of using a loss normalization factor.

We use SentencePiece tokenization [19] to generate a vocabulary of size 64,000. The corpus used to generate the vocabulary is sampled from the training data using temperature sampling with $\tau = 3.33$.

We use the T5X library [25] to train the models. For all experiments, we use the Adafactor optimizer [27], where we use momentum, and we do not factorize the second moment of the Adafactor states. The baseline run without fine-tuning, and the pre-training phase of our proposed method, was run with a constant learning rate of 0.01 in the first 10,000 steps and inverse square root decay afterwards. For the fine-tuning phase of our method, we reset the optimizer state, and do a 10,000-step linear warmup with inverse square root decay afterwards.

## C  Discussion on Sampling Rate Schedules

From our preliminary experiments on using schedules for the sampling rates in the NMT workloads, we find that the learning rate schedule must be tuned accordingly, which affects the overall performance of the run. For example, we find that cosine decay schedule performs better than inverse square root decay for scalarization. However, if we use cosine learning rate decay in conjunction with linear sampling rate decay (used by DDS, and defining sampling rate to be for the high-resource language-pair), by the time the sampling rate for low-resource task is high enough, the learning rate has decayed rapidly (by nature of cosine decay), resulting in little learning for the low-resource task. Using inverse square root learning rate decay solves this issue, but this results in overall worse performance due to the suboptimal learning rate schedule. In contrast, our method is free to use any scheduler that maximizes performance in each leg of training (pre-training and fine-tuning). Lastly, when tuning hyperparameters, using dynamic sampling rates requires executing the full training run many times. On the other hand, for our method, we can focus our resources on tuning the fine-tuning phase, (since the pre-training phase has only one task, and is an easier optimization problem) which is shorter than the total training time.

