# OpenReview forum: "Order Matters in the Presence of Dataset Imbalance for Multilingual Learning"
_NeurIPS.cc/2023/Conference — NeurIPS 2023 poster_

### Official Review · Reviewer_evVX · 2023-07-06

**Soundness:** 3 good
**Presentation:** 4 excellent
**Contribution:** 3 good
**Rating:** 7
**Confidence:** 4

**Summary:**

This paper presents a simple and effective multi-task learning strategy of a joint pretraining followed by fine-tuning, where pretraining is on the high-resource task and fine-tuning is on a mixture of high and low-resource tasks. This significantly improves performance on the low-resource tasks, while performing at par or even better sometimes on high-resource tasks. The authors show results on a machine translation task consisting of two high-resource and two low-resource language pairs and a language modeling task trained on five different languages.

**Strengths:**

- The proposed idea is attractive in its simplicity, and offers good validation loss reductions on translation and language modeling tasks.
- The authors provide a fairly extensive empirical analysis of the proposed approach, and offer good baselines to compare against.
- The paper is written clearly and the overall narrative flows well.

**Weaknesses:**

While the proposed approach has been empirically validated with multiple experiments, there remain unanswered questions about the initial pretraining on a high-resource language:
- How important is the pretraining objective?
- What is the influence of the chosen high-resource language on final performance on low-resource languages?
- Why does pretraining longer improve the loss for high-resource NMT tasks but degrade the loss for high-resource language modeling tasks (c.f., Figure 9(a) and 9(b)).

**Questions:**

- In Figure 9a, pretraining longer on English seems to benefit Gujarati but not Hindi. Any thoughts on why this might be?

- From line 260, "Longer pre-training helps more in terms of performance than exposure to the tokens of the low-resource task" -- This claim demands further probing. Is this dependent on the choice of high-resource language for pretraining? This claim holds for Gujarati but not Hindi. Are there features of the low-resource task (e.g., type-token ratio in the unlabeled text for the low-resource tasks) that makes transfer from the high-resource task more effective?

- In the many-task setting, as in Section 3.2, would it be beneficial to introduce an intermediate pretraining stage between pretraining and finetuning? Such strategies have been shown to be effective in prior work (e.g., Phang et al., " Sentence encoders on stilts: Supplementary training on intermediate labeled-data tasks", 2018). Also, such works on intermediate labeled-data tasks should be cited in the related work section.

- The authors should move up the BLEU results from the Appendix to the main draft. For NMT, it is important to show that validation loss reductions do translate to BLEU score improvements (especially for English->Hindi).

- A suggestion: Since pretraining is usually associated with self-supervised objectives, it would be useful to explicitly clarify that the pretraining and finetuning objectives are both cross-entropy losses for the NMT task.

**Limitations:**

Limitations have been adequately addressed.

---

> ### Author Rebuttal · Authors · 2023-08-10
>
> We thank the reviewer for their detailed feedback. We respond to the weaknesses and questions brought up below:
>
> **W:** *Why does pretraining longer improve the loss for high-resource NMT tasks but degrade the loss for high-resource language modeling tasks (c.f., Figure 9(a) and 9(b)).*
>
> **Response:** For all our experiments, when starting the fine-tuning phase of training we reset the cosine learning rate schedule with a warmup. Due to the warmup, the performance of the pre-trained high-resource task performance worsens a bit before improving again (see the panel corresponding to En for Figure 8). Because of this, the high-resource task needs enough training steps to recover its old performance, and to continue improving.
>
> In the language modeling experiment (Figure 9(a)), we kept the total training budget the same, which meant increasing pre-training length resulted in decreased fine-tuning length. Less fine-tuning time means less time to recover for En, hence the worse performance.
> On the other hand, for the NMT experiment (Figure 9(b)), we kept the fine-tuning length the same regardless of the pre-training length, which is why the En-Fr performance could keep improving.
>
> The implications of these results is that when there is a strict training budget, it’s better to be conservative and pre-train for a shorter amount of time. However, if the goal is to obtain the best performance and there is no strict compute budget, it’s better to pre-train for as long as possible before fine-tuning. Note that longer overall training is an option for our method (by pre-training for longer) but not for scalarization because scalarization needs to always be training on the low-resource tasks, which will lead to overfitting when training for long.
>
> We will update our final draft to be more clear about this.
>
> **Q1:** *In Figure 9a, pretraining longer on English seems to benefit Gujarati but not Hindi. Any thoughts on why this might be?*
>
> **Response:** In Figure 9a, as we pre-train for longer, we fine-tune for a shorter amount of time. In addition to this, Hindi is a high-resource task given our training budget. Therefore, overall lesser training for Hindi resulted in worse performance. Pre-training still benefits Hindi– Figure 8 shows that pre-training initialized Hi performance at a better position than random initialization, and resulted in more data-efficient training.
>
> As an aside, we did not include Hindi in the pre-training phase because we wanted to keep our method simple. Including Hindi in the pre-training phase would require figuring out good sampling proportions not only for the fine-tuning phase, but for the pre-training phase. We can avoid under-training high-resource tasks by fine-tuning for longer.
>
> **Q2:** *"Longer pre-training helps more in terms of performance than exposure to the tokens of the low-resource task" -- This claim demands further probing. Is this dependent on the choice of high-resource language for pretraining? This claim holds for Gujarati but not Hindi. Are there features of the low-resource task (e.g., type-token ratio in the unlabeled text for the low-resource tasks) that makes transfer from the high-resource task more effective?*
>
> **Response:** We hope our response to Q1 helped answer a part of this question. In general, the relationship between the pre-training and fine-tuning tasks should have an impact on performance, although we are unsure what exactly about the relationship causes better transfer. We believe this is an important question that deserves separate investigation. In our work, we stuck to the simplest method and chose pre-training tasks (En-Fr for MT and En for language modeling) that are (and will always be) available at an abundance, therefore, are perfect to pre-train on.
>
> **Q3:** *In the many-task setting, as in Section 3.2, would it be beneficial to introduce an intermediate pretraining stage between pretraining and finetuning? Such strategies have been shown to be effective in prior work (e.g., Phang et al., " Sentence encoders on stilts: Supplementary training on intermediate labeled-data tasks", 2018). Also, such works on intermediate labeled-data tasks should be cited in the related work section.*
>
> **Response:** Thank you for the missed citation– we will include intermediate pre-training works in the related work section of our final draft.
>
> We have not tested with intermediate training in our context. In Phang et al., Intermediate training on a different but related task seemed to help in the monolingual (English) setting, which means it could possibly improve our method further. In our paper, we focus on cross-lingual transfer, but we believe that pre-training to utilize cross-task transfer is an interesting question to be studied in the future.
>
> **Q4:** *The authors should move up the BLEU results from the Appendix to the main draft.*
>
> **Response:** Thank you for the suggestion, we will move the BLEU results to the main section in our final draft.
>
> **Q5:** *A suggestion: Since pretraining is usually associated with self-supervised objectives, it would be useful to explicitly clarify that the pretraining and finetuning objectives are both cross-entropy losses for the NMT task.*
>
> **Response:** Thank you for the suggestion, we will clarify this in the final draft.
>
> We hope that we clarified some of the reviewer’s concerns– we will update our final draft to make our points more clear. Please let us know if there are any further questions.

---

> > ### Comment · Reviewer_evVX · 2023-08-19
> >
> > Thanks to the authors for their detailed responses.
> >
> > > The implications of these results is that when there is a strict training budget, it’s better to be conservative and pre-train for a shorter amount of time. However, if the goal is to obtain the best performance and there is no strict compute budget, it’s better to pre-train for as long as possible before fine-tuning.
> >
> > I think this is an important point that the authors should explicitly highlight in the main draft.
> >
> > I am raising my score to an Accept.

---

### Official Review · Reviewer_V1qx · 2023-07-06

**Soundness:** 3 good
**Presentation:** 3 good
**Contribution:** 2 fair
**Rating:** 5
**Confidence:** 4

**Summary:**

The paper proposes a method called pretraining and joint-finetuning, which pretrains a model on a high-resource task than finetunes the model with joint high- and low-resource tasks, benefiting from both static sampling method and naïve transfer learning. The method is verified on multilingual translation and multilingual language modelling. Experiments show improvement on the two scenarios in term of validation loss.

**Strengths:**

1. The proposed method is simple and easy to implement/reproduce.
2. The method can achieve lower validation loss compared to static sampling and naïve transfer learning.
3. The method verified on multilingual (> 2) setting and extremely resource-imbalanced (English is 16745 times larger than the smallest language) setting.

**Weaknesses:**

1. The algorithm is sensitive to the sampling proportion, resulting a need to search proportion for every model trained. In addition to searching the proportion of dataset size, there is a need for grid search of N1 and N2 (N1 + N2 = N). The hyperparameters (proportion = 0.4 for two tasks, tau = 3.33 for > 2 tasks, and joint-finetuning steps = 50k) are not verified among different tasks.
2. Experiments are done only on multilingual setting, whose scope is a bit different from the title “multitask” learning. Is the proposed method still effective on multitask like NER + POS + sentiment analysis?
3. The improvement on performance is not significant when the evaluation metric is BLUE (from Fig. 26).

**Questions:**

1. Definition of Pareto optimal points. According to the paper, “θ is Pareto optimal if it is not dominated by any other point”. However, for a parameter family for a certain architecture, it is impossible to test all parameters in the whole space. In this case, is it rigorous to say that a parameter θ is not dominated by any other parameters?
2. Inadequate references. There are plenty of papers on the catastrophic forgetting phenomenon in multilingual/multitask learning. I think the claims in this paper might been seen from others. For example, the “order matters” claim has been found in “Overcoming Catastrophic Forgetting beyond Continual Learning: Balanced Training for Neural Machine Translation”. The authors should check previous literature more carefully.
3. Scalarization implementation. In Line 72 of the paper, the static sampling is implementing by adding weights to the losses from different tasks (languages). Will the models trained with different proportions be fed different data? For example, suppose the En->Ro data is fixed, will a model with 0.1 En-Ro proportion be fed more En data than a model with 0.9 En-Ro proportion? If the answer is no (i.e., the data is the same, the only difference is the weights on losses), does the method have a same effect with sampling data in each batch?
4. Why the Figures do not always show all results of the ten proportions (e.g., Fig. 2 Left)?



Minor suggestions (mainly for presentation):
- I think having a separate section about the proposed method will be better.
- For figures of the loss that need a comparison (e.g., Fig. 3 and 8), the illustration will be clearer if the scale of the two subfigures is consistent.



After Author Response:
I strongly encourage the authors to address Questions 1 and 2, specifically the clarification of Pareto dominance and a more comprehensive discussion with previous literature. These enhancements would undoubtedly contribute to the rigor and solidity of the paper.

**Limitations:**

The authors consider an interesting possibility of multiple high-resource languages setting as a future work, which can be seen as a partition of tasks. However, they do not provide a set of hyperparameters effective across tasks, e.g., at least how many steps of joint-finetuning should be taken? Which sampling proportion should we use when training with new data/tasks?

---

> ### Author Rebuttal · Authors · 2023-08-10
>
> We thank the reviewer for their feedback. Here is our response:
>
> **W1a:** *The algorithm is sensitive to the sampling proportion, resulting a need to search proportion for every model trained.*
>
> **Response:**
> Our method is not any more sensitive to the sampling proportion than scalarization, which is our baseline. Furthermore, the best sampling proportions will always depend on the task at hand. Therefore, even for scalarization, ideally one would search for a good sampling proportion.
>
> In our experiments, we follow two different schemes for testing sampling rates depending on what we aim to achieve. In our NMT experiments, the purpose of testing sampling rates in a grid was to investigate whether pre-training can improve the Pareto front of scalarization. We emphasize that we tested sampling rates in a grid for the scalarization baseline as well, so that the comparison is fair. In our language modeling experiments, our goal was to model reality as much as possible (by using temperature sampling), while still comparing against the best baseline. This is why we **tuned the temperature parameter for the scalarization baseline**, and used that same temperature for our method.
>
> **W1b:** *In addition to searching the proportion of dataset size, there is a need for grid search of N1 and N2 (N1 + N2 = N).*
>
> **Response:** Our method does indeed introduce an extra parameter to decide on compared to scalarization: scalarization only needs to decide on N, but we need to decide on N1 and N2. Regarding this, we would like to emphasize 3 things:
> - **We did not tune or perform a grid search in our experiments.** We chose values that were either the most beneficial to our baseline (scalarization), or seemed reasonable, and we continued using them without further tuning them.
> - **Parameters N1 and N2 are precisely how our method can do early stopping.** Let’s imagine using scalarization, and one of the tasks starts overfitting. It’s not ideal to do early stopping since the high-resource tasks can benefit from training longer. If we instead pre-train on the high-resource task, we can get a head-start on the high-resource task such that once the low-resource tasks start overfitting, we can stop training.
> - **Based on our experiments, we can recommend how one would decide on N1 and N2.** Our results show that pre-training doesn’t hurt performance, and the longer we pre-train for, the better performance we get. This means that if a practitioner is strictly bound with a compute budget, they should probably pre-train for only a fraction of their total budget to make sure that all tasks are able to converge. On the other hand, for a practitioner who mostly cares about best performance achievable, and does not have a strict computational budget, they should pre-train for as long as possible before joint fine-tuning.
>
> **W1c:** *The hyperparameters (proportion = 0.4 for two tasks, tau = 3.33 for > 2 tasks, and joint-finetuning steps = 50k) are not verified among different tasks.*
>
> **Response:** The performance of any given sampling rate depends on the task. Therefore, just like how it is unreasonable to believe that a single temperature parameter should be recommended for scalarization, we do not recommend a single proportion/temperature to be used for different tasks. We emphasize that we did not tune the sampling rate any more than the baseline scalarization.
>
> The number of steps to fine-tune also depends on the task. Some tasks will take only 50k steps to converge (En-Ro, or En-Hi), while some tasks will take longer (as in the language modeling experiments).
>
> **W2:** *Experiments are done only on multilingual setting, whose scope is a bit different from the title “multitask” learning. Is the proposed method still effective on multitask like NER + POS + sentiment analysis?*
>
> **Response:** Please see to our global response.
>
> **W3:** *The improvement on performance is not significant when the evaluation metric is BLUE (from Fig. 26).*
>
> **Response:** Our improvement on En-{Ro, Fr} is indeed weak in terms of the BLEU score. The improvements are much better for En-{Hi, Fr}, so we expect the amount of improvement to vary depending on the task.
>
> **Q1:** *Is it rigorous to say that a parameter $\theta$ is not dominated by any other parameters?*
>
> **Response:** You are right– we cannot rigorously guarantee that a given point is not dominated. We will update our writing to be more clear about this. However, we do believe that we did a reasonably good job of approximating the Pareto front by testing a grid of sampling rates and learning rates.
>
> **Q2:** *Inadequate references.*
>
> **Response:** Thank you for pointing this out, we will update our paper with more references on catastrophic forgetting.
>
> **Q3:** *Scalarization implementation. In Line 72 of the paper, the static sampling is implementing by adding weights to the losses from different tasks (languages). Will the models trained with different proportions be fed different data?*
>
> **Response:** We apologize for the confusion. We follow the convention in the NMT literature and implement scalarization via proportional sampling, where the average number of data corresponding to task i in a batch is proportional to w_i. So we do not change the loss function, only the way we sample the data. We will update our draft to include this point, thank you for pointing it out.
>
> **Q4:** *Why the Figures do not always show all results of the ten proportions (e.g., Fig. 2 Left)?*
>
> **Response:** This is because we are only plotting the points that were not dominated (considering only the space of parameters obtained through our grid searches). In the existence of low-resource tasks, the validation loss plot does not mirror the training loss plot like in the high-high case (Figure 1), and there are certain sampling rates that are more optimal than others.
>
> Regarding the minor suggestions, thank you for the feedback– we will update our draft to reflect them.

---

> > ### Comment · Area_Chair_Sn9W · 2023-08-18
> >
> > Reviewer V1qx, could you please review the latest response of the authors and indicate whether it addresses your feedback and whether you choose to maintain or change your rating?

---

> > ### Comment · Reviewer_V1qx · 2023-08-20
> >
> > Thank you for the detailed explanation provided within the response, particularly in regard to parameter selection. Based on the improvements and clarifications, I am prepared to endorse the acceptance of this paper, increasing my score from 4 to 5. However, I strongly encourage the authors to address Questions 1 and 2, specifically the clarification of Pareto dominance and a more comprehensive discussion with previous literature. These enhancements would undoubtedly contribute to the rigor and solidity of the paper.

---

### Official Review · Reviewer_9ZBe · 2023-07-07

**Soundness:** 3 good
**Presentation:** 4 excellent
**Contribution:** 3 good
**Rating:** 7
**Confidence:** 4

**Summary:**

The paper presents a multi-task training approach that involves pre-training on high resource tasks, followed by joint fine-tuning on the full set of tasks. The intuition is that high-resource tasks need a larger number of steps to reach convergence, whereas low-resource tasks will risk over-fitting if trained for that same number of steps.

The authors show that the proposed approach improves the performance for neural machine translation of low-resource languages (en-ro and en-hi, pretrained on en-fr) and similarly improves performance for multilingual language modelling for the case of low-resource languages.

**Strengths:**

* The authors show that _pretraining joint-finetuning_ is a simple approach to boost low-resource task performance in multi-task training. The experiments, which focus on language modelling and machine translation, are clearly designed and presented, with reasonable baselines.
* In this proposed training approach, the pre-training phase ends up taking up most of the training time. Since pre-training is conducted on the subset of high-resource tasks, this makes it an easier optimisation problem compared to the standard approach where all tasks are trained. The potentially trickier joint finetuning step requires overall fewer steps. Hyperparameter sweeps could plausibly be limited just to this final step, making them cheaper and faster compared to the baseline joint training approach.
* Between the paper and the appendices, enough details are given so that reproducing these run should present no issues.

**Weaknesses:**

* I would have loved to see some more realistic MT experimental settings. Pretraining on en-fr for either en-zh or en-hi (or viceversa) feels very unrealistic. There are much closer language pairs that could have been used, which would have made the results more convincing.
* I am concerned about the results of the experiments of section 3.1.1. What's the effect on BLEU scores here? It would have been good to include these in Appendix A5, as was done for the other two language pairs.

Minor:
* Line 61: _if_ is attached to $\forall$
* Line 171: shows -> show
* Nitpick: In the abstract, the wording "experimental settings which range from neural machine translation (NMT) to multi-lingual language modeling" seems a bit out of place. These are exactly the two settings on which the proposed approach is being evaluated, there aren't any other experimental settings that fall within these two.

**Questions:**

* Line 128 refers the reader to Figures 21 and 22 in the appendix to show that the proposed approach does not hurt performance in the high-high case. It's not clear to me how the said figures show this. The clearest evidence for me would have been a version of Figures 26 and 27 for the English+Chinese case. Would you be able to provide such a plot?

**Limitations:**

The authors do a good job of listing and addressing limitations.

---

> ### Author Rebuttal · Authors · 2023-08-10
>
> We thank the reviewer for their detailed feedback and positive response. Our responses to the weaknesses and questions brought up are below:
>
> **W1:** *I would have loved to see some more realistic MT experimental settings. Pretraining on en-fr for either en-zh or en-hi (or vice versa) feels very unrealistic. There are much closer language pairs that could have been used, which would have made the results more convincing.*
>
> **Response:** In our MT experiments, we tried to choose task pairs with varying degrees of transfer-ability from En-Fr (Ro is more semantically closer to Fr than Hi and Zh are to Fr). If we only chose pairs that transferred well, it would be unclear whether our method showed improvement only because of the transferability, and whether it will work in settings where the tasks are not similar. We do agree that having more experiments in the realistic setting would make our arguments stronger.
>
> **W2:** *I am concerned about the results of the experiments of section 3.1.1. What's the effect on BLEU scores here?*
>
> **Response:** The key point we emphasize in section 3.1.1 is that for a high-resource task, the best performance achievable is bottlenecked by the amount of data seen for the task. The best En-Zh performance for pre-training (300k) joint fine-tuning (300k) is worse than the best En-Zh performance for joint training (600k) since the latter sees more En-Zh data than is possible for the former. At the same time, the best En-Zh performance for pre-training (300k) joint fine-tuning (300k) is equal to the best En-Zh performance seen by joint training (300k) which saw the same amount of En-Zh data.
>
> The implications of these results is that when there is abundant data available for a task, and overfitting is not an issue, it is important to train on it as much as possible, and that pre-training on a different task cannot act as a proxy for more data on that task.
>
> Lastly, we generated the BLEU score plot for En-{Zh,Fr} (see pdf attached in our global response). From the figure, we can see that the BLEU score version shows similar conclusions as the cross-entropy version, but with a more positive result– best performance for En-Zh is on-par with joint training for 600k steps.
>
> **Q:** *Line 128 refers the reader to Figures 21 and 22 in the appendix to show that the proposed approach does not hurt performance in the high-high case. It's not clear to me how the said figures show this.*
>
> **Response:** We apologize for the confusion. When we said the proposed approach does not hurt performance, we meant that our method does not worsen the best possible En-Zh performance achievable **given the same amount of En-Zh data** available for the model to see.
>
> Elaborating on our response to W2, let’s imagine the Pareto front of pre-training on En–Fr for 300k steps and then joint training on En-{Zh, Fr} for 300k steps, and compare it to just joint training for 300k steps. The pareto front should be improved for the En-Fr task performance since it saw a lot more of the En-Fr task data. For the En-Zh task performance, we can expect one of 3 possibilities:
> 1. The front is improved due to positive transfer despite seeing the same amount of En-Zh data
> 2. The front is unchanged
> 3. The front is worsened due to pre-training on En-Fr possibly resulting in worse initialization and/or slow training for En-Zh.
>
> Figure 1 shows that case 2 is indeed what happens– the best En-Zh performance for pre-training (300k) joint fine-tuning (300k) is equal to the best En-Zh performance seen by joint training (300k). Pre-training on En-Fr cannot act as a proxy for more En-Zh data, because if it could, then the front would be improved. At the same time, pre-training also does not negatively impact En-Zh training– Figures 21 and 22 show that pre-training does not affect the learning efficiency for En-Zh (slope of the curves are similar to one another), and also did not result in a worse initialization for En->Zh.
>
> We hope that our response and our newly generated BLEU score plot (see pdf in global author response) addressed some of the reviewer’s concerns. We will update our writing for section 3.1.1 to make this point more clear.
>
> **Regarding the minor fixes**, thank you for the suggestion and for catching the typos– we will make the changes to the final version.

---

> > ### Comment · Reviewer_9ZBe · 2023-08-21
> >
> > Thank you for the response. Having read this as well as other reviews and rebuttals, I confirm my assessment of the paper.

---

### Official Review · Reviewer_wUUc · 2023-07-09

**Soundness:** 4 excellent
**Presentation:** 4 excellent
**Contribution:** 3 good
**Rating:** 6
**Confidence:** 4

**Summary:**

This paper presents an analysis of multi-task training with the existence of low-resource tasks (in the context of the paper, they focused on multi languages). The paper argues that it is better to first "pre-train" on a high resource task and then "fine-tune" on a joint of high and low resource tasks. Experiments were conducted in neural machine translation and multilingual language modeling to support the hypothesis.

**Strengths:**

The proposed hypothesis intuitively makes sense and the experiments have been conducted in a careful setting. The proposed solution is simple, but has its effectiveness verified in NMT and language modeling. The paper has a smooth flow and is easy to understand.

**Weaknesses:**

1. While the proposed solution is effective, the benefits of having a multilingual model (or in general a multi-task model) is that a single model can be used in various situations, without the need of (continue) training on different data splits and producing several models. I feel at the current stage the application of this finding might be limited.

**Questions:**

The paper focuses on a specific multi-task setup of multilinguality, however, the finding/hypothesis doesn't seem to have strong ties with multilingualityThe paper presented some limitations on scaling the tasks and settings to be analyzed., and is therefore not necessarily confined to this setting. I am curious if the authors conducted any experiments on the the general multi-task setup.

**Limitations:**

The paper presented some limitations on scaling the tasks and settings to be analyzed.

---

> ### Author Rebuttal · Authors · 2023-08-10
>
> We thank the reviewer for their feedback. Our responses to weaknesses and questions are below:
>
> **W:** *[…]the benefits of having a multilingual model (or in general a multi-task model) is that a single model can be used in various situations, without the need of (continue) training on different data splits and producing several models.*
>
> **Response:** We completely agree with the reviewer about the benefits of multitask models, and how one model can be used in various situations. We believe that our proposed method does not deviate from this. Our method can be thought of as a specific scheduling strategy for sampling rates to balance the tasks. While the current convention is to train on all the tasks at once with a fixed ratio, we propose to use a 100% sampling rate for high-resource tasks in the beginning of training, before training on all the tasks with a fixed ratio. Our method yields a single model that can be used on all tasks trained, and does not need to be trained further on different data splits.
>
> **Q:** *I am curious if the authors conducted any experiments on the general multi-task setup.*
>
> **Response:** In our work, we frame multilingual learning as a multi-task optimization problem (as in many prior work [1, 2, 3, 4, 5]), and focus on utilizing cross-lingual transfer. We have yet to conduct experiments on the “general multi-task setup” (multiple NLP tasks that may or may not be multilingual), where the focus would be on utilizing cross-task transfer. We will update our future work section to reflect this.
>
> [1] Luong et al, 2015, Multi-task Sequence to Sequence Learning
>
> [2] Firat et al, 2016, Multi-Way, Multilingual Neural Machine Translation with a Shared Attention Mechanism
>
> [3] Arivazhagan et al, 2019, Massively Multilingual Neural Machine Translation in the Wild: Findings and Challenges
>
> [4] Jean et al, 2019, Adaptive Scheduling for Multi-Task Learning
>
> [5] Wang et al, 2020, Gradient Vaccine: Investigating and Improving Multi-task Optimization in Massively Multilingual Models

---

> > ### Comment · Reviewer_wUUc · 2023-08-16
> >
> > Thanks for the clarifications! I now believe that the finding of this paper is worth exploring on a wider spectrum of setups, including multi-task learning, or multilingal pre-training besides NMT. I am increasing my score from 5 to 6.

---

### Official Review · Reviewer_yZ9n · 2023-07-09

**Soundness:** 4 excellent
**Presentation:** 3 good
**Contribution:** 3 good
**Rating:** 6
**Confidence:** 3

**Summary:**

This paper presents an empirical finding: in the context of data imbalance in multi-task learning, pre-training on high-resource tasks then fine-tuning on a mixture of high/low-resource tasks can achieves superior results, compared to standard weighted sampling training. Authors applied the proposed training method *pre-training joint fine-tuning* to neural machine translation (NMT) and multi-lingual language modeling. Both experimental results shows that the proposed method allows the high-resource task to converge faster while preventing overfitting in low-resource tasks.

**Strengths:**

- Achieving a good balance between low-resource and high-resource languages during multitask training is a critical research area, particularly in the context of large-scale language modeling. The proposed approach of pre-training followed by joint fine-tuning is a timely solution that addresses this challenge.
- The results obtained from the empirical study demonstrate the superiority of the pre-training joint fine-tuning approach over the conventional method of weighted sampling during training. The model trained using this approach consistently outperforms existing methods, showcasing its effectiveness in addressing the challenges posed by data imbalance in multi-task learning.
- An additional ablation study showed that the pre-training joint fine-tuning has a regularization effect (to avoid overfiting to low resource language). This improvement cannot be achieved by simply increasing drop-out ratio.

**Weaknesses:**

- In the main paper, authors only reported perplexity metrics in the evaluation results. For multi-lingual language modeling task, I would like to see the evaluation on downstream tasks as in the mT5 paper. It would be helpful to demonstrate the impact of the proposed method if authors can show the improvement in downstream tasks on XQuAD or XTREME benchmark.

**Questions:**

Why temperature based sampling is only applied in the language modeling task not the NMT task? What is the advantage of temperature based sampling compared to scalarization in the large scale dataset?

**Limitations:**

I am curious to see what if the authors applied this method in a more realistic setting: training LM on the whole mC4 dataset, and then evaluate the performance (low-resource language vs high-resource language) on different downstream tasks as in mT5.

---

> ### Author Rebuttal · Authors · 2023-08-10
>
> We thank the reviewer for their feedback. Here is our response to the weaknesses and questions pointed out by the reviewer.
>
> **W1a:** *In the main paper, authors only reported perplexity metrics in the evaluation results.*
>
> **Response:** For the NMT experiments, we report BLEU score evaluations in Appendix A.5.
>
>
> **W1b:** *For multi-lingual language modeling task, I would like to see the evaluation on downstream tasks as in the mT5 paper.*
>
> **Response:** We agree with the reviewer on the importance of evaluating our proposed method on downstream tasks. We did not evaluate on downstream tasks for our paper because almost all languages included in the downstream tasks for evaluating cross-lingual generalization ability (e.g. XTREME) are high-resource in mC4 given our training budget (with the exception of Swahili and Yoruba). Furthermore, we believe that it will be more meaningful to evaluate on such downstream tasks after pre-training on many more languages for a longer time. We leave the scaling up of our method for future work to study, and we will expand on our limitations section to reflect on the reviewer’s comment.
>
> **Q1:** *Why temperature based sampling is only applied in the language modeling task not the NMT task?*
>
> **Response:** We do not need to do temperature sampling for the NMT tasks because we already search through the possible sampling rates in a grid. In other words, the sampling rates we have tried for NMT already include temperature sampling.
>
> In the NMT experiments, we only have two tasks (language-pairs), which allows us to test sampling ratios in a grid (e.g. For En->{A, B}, we can try sampling rate x for En->A and 1-x for En->B, where x is in {0.1, 0.2, …, 0.9}). Doing so for more than 2 tasks will require testing exponentially many sampling ratios, which is why for the 5 task case in the language modeling experiment, we resort to temperature sampling.
>
> **Q2:** *What is the advantage of temperature based sampling compared to scalarization in the large scale dataset?*
>
> **Response:** Temperature sampling is a heuristic to obtain sampling rates to be used for scalarization (since for higher number of tasks, testing a grid of sampling rates is not feasible). Many prior work used temperature sampling with various temperatures, and its advantage is that it is simple, intuitive, and can be controlled with a single parameter.

---

### Author Rebuttal · Authors · 2023-08-10

**Q:** *Reviewers wUUC and V1qx both pointed out that our method focused on the multilingual setup, and were curious whether we ran experiments on the general multi-task setup.*

**Response**: In our work, we frame multilingual learning (NMT, language modeling) as a multi-task optimization problem as in many prior work [1, 2, 3, 4, 5]. Using translation as an example, learning to translate to/from multiple languages, and learning multiple “tasks” (e.g. question answering, and summarization) can both be seen as learning multiple functions at once.

Since our work focuses on utilizing cross-lingual transfer, we have yet to conduct experiments on the “general multi-task setup” (multiple NLP tasks that may or may not be multilingual), where the focus would be on utilizing cross-task transfer. We will update our future work section to reflect this.

[1] Luong et al, 2015, Multi-task Sequence to Sequence Learning

[2] Firat et al, 2016, Multi-Way, Multilingual Neural Machine Translation with a Shared Attention Mechanism

[3] Arivazhagan et al, 2019, Massively Multilingual Neural Machine Translation in the Wild: Findings and Challenges

[4] Jean et al, 2019, Adaptive Scheduling for Multi-Task Learning

[5] Wang et al, 2020, Gradient Vaccine: Investigating and Improving Multi-task Optimization in Massively Multilingual Models

**Lastly**, we attach BLEU score plots for En-{Zh,Fr} as requested by Reviewer 9ZBe.

---

### Decision · Program_Chairs · 2023-09-21

**Decision:**

Accept (poster)

**Comment:**

This paper investigates the strategies on how to best deal with data imbalance in multilingual learning. Their proposed method enables high-resource tasks to converge faster and prevents overfitting for low-resource tasks. The method is clearly described and evaluated on both NMT and language modeling. The paper would benefit from a more comprehensive discussion of prior work, further clarification of pareto dominance, additional evaluation on XQuAD or XTREME as pointed out by reviewers, and evaluation on a wider range of language for NMT.